# Behavior change and infection induced immunity led to the decline of the 2022 Mpox outbreak in Berlin

Nils Gubela [1,2] ✉, Hee-yeong Kim [3], Nikolay Lunchenkov [4,5], Daniel Stern [6], Janine Michel [7], Andreas Nitsche[7], Axel J. Schmidt [8,9], Ulrich Marcus [4] & Max von Kleist [1,3]

## Abstract

**Background** Mpox denotes a viral zoonosis caused by the Orthopoxvirus monkeypox (MPXV), which is endemic in West and Central Africa. In spring 2022, notable outbreaks of MPXV clade IIb were recorded in several high-income countries, predominantly affecting men who have sex with men (MSM). At the peak of the outbreak, over 200 new mpox cases per week were reported in Berlin, which constitutes one of the largest MSM population in Europe. Within the same year, the outbreak significantly declined, and it is unclear which factors contributed to this rapid decrease.
**Methods** To investigate the concomitant effects of sexual contact networks, transient contact reductions and the effect of infection- vs. vaccine-derived immunity on the 2022 mpox outbreak, we calibrated an agent-based model with epidemic, vaccination, contact- and behavioral data.
**Results** Our results indicate that vaccination has a marginal effect on the epidemic decline. Rather, a combination of infection-induced immunity of high-contact individuals, as well as transient behavior changes reduce the number of susceptible individuals below the epidemic threshold. However, the 2022 mpox vaccination campaign, together with infection-derived immunity may contribute to herd-immunity in the Berlin MSM population against ongoing clade I mpox outbreaks. Demographic changes and immune waning may deteriorate this herd immunity over time.
**Conclusions** These findings highlight that, in addition to vaccination, timely and clear communication of transmission routes may trigger spontaneous protective behavior within key populations; underscoring the importance of targeted sexual health education as a core component of outbreak response.

## Plain language summary

Mpox is a virus disease that is transmitted through direct contact with infected individuals or contaminated materials. In 2022, major Mpox outbreaks occurred in several countries, especially among men who have sex with men (MSM). At its peak, Berlin (Germany) reported over 200 new cases per week, but saw a rapid decline of cases during summer 2022. We used mathematical simulations to explore how patterns of sexual contact, temporary changes in behavior, and immunity from infection or vaccination may have impacted the outbreak. We found that vaccination had a minor effect, while immunity from past infections and behavior changes played a major role in reducing infections. These insights show that clear communication about virus spread can encourage protective behaviors, highlighting the importance of targeted sexual health education in containing sexually transmitted infections.

The monkeypox virus (MPXV) was first discovered in humans in the Democratic Republic of Congo in 1970[1] and has since caused several outbreaks of human mpox. MPXV clade I mainly circulates in Central Africa, and clade II is predominantly found in Western Africa. In May 2022, a global mpox outbreak with MPXV Clade IIb occured, causing over 102,000 laboratory confirmed cases[2]. This outbreak spread to over 100 countries[3], prompting the WHO to declare it a public health emergency of international concern on July 23, 2022[4]. Several distinctive features set this outbreak apart

[1]Department of Mathematics & Computer Science, Freie Universität Berlin, Berlin, Germany. [2]International Max-Planck Research School for Biology and Computation (IMPRS-BAC), Max-Planck Institute for Molecular Genetics, Berlin, Germany. [3]Project Group 5 "Systems Medicine of Infectious Disease", Robert Koch Institute, Berlin, Germany. [4]Department of Infectious Disease Epidemiology, Robert Koch Institute, Berlin, Germany. [5]TUM School of Social Sciences and Technology, Technical University of Munich, Munich, Germany. [6]Centre for Biological Threats and Special Pathogens, Biological Toxins (ZBS3), Robert Koch Institute, Berlin, Germany. [7]Centre for Biological Threats and Special Pathogens, Highly Pathogenic Viruses (ZBS1), German Consultant Laboratory for Poxviruses, WHO Collaboration Center for Emerging Threats and Special Pathogens, Berlin, Germany. [8]Deutsche Aidshilfe, Berlin, Germany. [9]Sigma Research, Department of Social and Environmental Health Research, London School of Hygiene & Tropical Medicine, London, UK. ✉e-mail: nils.gubela@fu-berlin.de

from previous human mpox outbreaks, namely most patients were in their thirties, male, and pathological presentation was primarily through ano-genital lesions, likely obtained through sexual contact with other men[5–7].

In Germany, a total of 4139 mpox cases have been reported to date, with the majority (3677 cases) occurring between May 2022 and autumn 2022[8]. Berlin, which harbors the largest population of men who have sex with men (MSM) in Germany[9], accounts for the majority of cases of any federal German state. Of the 1816 Berlin cases reported by January 2025, around 1600 cases occurred during the 2022 outbreak, predominantly among men who have sex with men (MSM), in particular gay men[10]. Case numbers rose until mid-June 2022, with only a few cases reported by October 2022, and none recorded between January 2023 and July 2023. Since August 2023, low levels of mpox cases have occasionally been reported with 56% of notifications sourced in Berlin and 67% with traced acquisition in Berlin (11% were imported cases from outside Berlin)[11]. Since October 2024, seven cases of imported clade Ib have been reported in Germany in individuals with a travel history to affected countries, with three secondary cases from the same household[12].

As part of the emergency response to the mpox 2022 outbreak, Modified Vaccinia Virus Ankara (MVA)-based vaccines were offered starting in July 2022, particularly to the MSM population. By the end of October 2022, slightly more than 15,000 first-dose vaccinations and 4300 second-dose vaccinations had been administered in Berlin, while the estimated number of self-identified gay men living in Berlin is ~60,000[13]. While the vaccine is known to induce a robust and protective immune response against mpox[14], neither infection nor vaccination provides complete protection against reinfection[15,16].

Different reasons may explain the decline of viral circulation, such as the buildup of population immunity in groups with high numbers of partners[17,18], a behavior change in MSM populations in anticipation of infection risks[19–22], the impact of the vaccination campaign[23], or a combination.

We calibrated a coupled within-host virus dynamics and agent-based epidemiological model to identify the factors that contributed to the decline of mpox cases in Berlin. The model was utilized to estimate the level of protective immunity within the MSM population against a potential new mpox outbreak. We show that vaccination had little to no effect on the decline of cases during the summer of 2022. Instead, a combination of transient contact changes and a depletion of high-contact individuals, who can sustain long infection chains, caused the decline in mpox cases. We estimate that the acquired immunity may reduce the probability of a new large outbreak, though it may deteriorate over time due to demographic changes and immune waning.

## Methods
### Epidemiological data
Mpox case numbers were obtained from the German reporting system[8] and the vaccination timeline from the German vaccine monitoring system at Robert Koch Institute[24]. The case numbers are reported as cases per calendar week. The first cases are reported in calendar week 20 (starting May 16, 2022). We include 2 weeks with no reported cases at the beginning and conclude observations with the last week of October 2022.

Vaccination data is provided monthly. We approximate the weekly vaccination numbers by dividing the monthly totals by the number of calendar weeks in that month. We assume that vaccine effects (vaccine efficacy and return to prepandemic contact behavior) manifest 2 weeks after vaccination[25]. Therefore, the effective vaccination timeline is shifted by 2 weeks (see Supplementary Fig. 1).

In Berlin, anonymous STI testing is offered outside of the primary care system in three inclusive community-based voluntary counselling and testing (CBVCT) centers. The attending clients are usually asymptomatic. Upon arrival in the CBVCT centers, the clients are asked to complete an anonymous questionnaire on sociodemographic, sexual behavior and sexual health topics[26]. In addition, rectal, urethral and pharyngeal swabs (separate or pooled), or urine samples are taken for gonorrhoea and

chlamydia screening. Pooled samples are combined rectal, urethral, urine, and/or pharyngeal sample of one person. To provide evidence of behavioral changes, we also report bacterial STI diagnosis rates from routine STI testing at three Berlin CBVCT centers from April to November 2022. These clinics offer low-cost HIV and STI testing, including urine and anal swab testing for gonorrhoea and chlamydia, primarily for asymptomatic gay/homosexual clients[27]. We received the number of positive tests for chlamydia or gonorrhea from CBVCT centers and we did not collect or analyze patient samples for this study. Every detection of gonorrhea is notifiable under the German Public Health Act (Infektionsschutzgesetz, IfSG) and must be reported to the Robert Koch Institute without requiring separate medical ethical clearance. v

### Population model
The number of gay men living in Berlins (~60,000) was derived from estimates based on the European MSM Internet Survey 2017[13]. To initialize the agents within our model, we utilized survey data from ref. 28, dividing the sample into vaccinated and unvaccinated subpopulations based on whether individuals had received their first dose of the vaccine.

For each subpopulation, we fitted an exponential distribution to the self-reported number of sexual partners, enabling us to represent the variability in partner numbers within each group accurately (see Supplementary Figs. 2–4). In 2022, a total of 18,104 first doses of the vaccine were administered, which provided an estimate for the size of the vaccinated MSM population. From this vaccinated subpopulation, we sampled 18,104 agents based on their degree (i.e., number of condomless anal sex partners) and assigned a behavior change status accordingly.

The remaining agents were sampled from the unvaccinated subpopulation as characterized in the survey. For these agents, we extracted data on the number of condomless anal sexual partners over a 3-month period, their vaccination status, and a categorical variable indicating the degree to which they altered their behavior during the 2022 mpox outbreak. This behavior change was classified as "strongly," "somewhat," or not changed. The contact distribution is shown in Fig. 1e.

### Network contact model with diseases spreading and progression
We model the spread of mpox in Berlin by representing sexual contacts as a temporal network and allowing the infection to propagate through it. Each agent has two network parameters, $\lambda_i^+$ and $\lambda_i^-$, which determine the rate with which new connections are formed or existing ones are broken, respectively. An edge between agent $i$ and $j$ is created with rate $\lambda_{ij}^+ = \lambda_i^+ \lambda_j^+$ and is removed with rate $\lambda_{ij}^- = 1$. The mpox virus spreads along a connection between a susceptible and an infected agent at a rate $\lambda^{\mathrm{inf}}$. Once exposed, a susceptible agent undergoes an incubation period before becoming infectious (Fig. 1a), at which point the agent can transmit the infection to connected susceptible agents.

To accurately replicate the within-host time course of infection progression[29], we employ a model with five infectious compartments (Fig. 1c). Infectious agents transition to a diagnosed state at a rate $\lambda^{\mathrm{diag}}$ (see Supplementary Fig. 5). Upon diagnosis, agents are isolated-they lose all existing contacts and refrain from forming new ones for the duration of their infectious period. After progressing through all infectious stages, agents recover. Recovered agents are not infectious and are immune to further infection for the remainder of the simulation.

Vaccination is administered at the start of each calendar week, based on a predetermined vaccination timeline. An agent is eligible for vaccination if, as indicated in the survey, the agent has been identified as vaccinated and has not yet been diagnosed with mpox. We assume an 80% vaccine efficacy for susceptible agents 14 days post vaccination[25], with the vaccine having no effect on agents already infected at the time of administration.

### Transient behavior change
Agents who reported modifying their behavior during the 2022 mpox outbreak in anticipation of infection reduce their expected number of

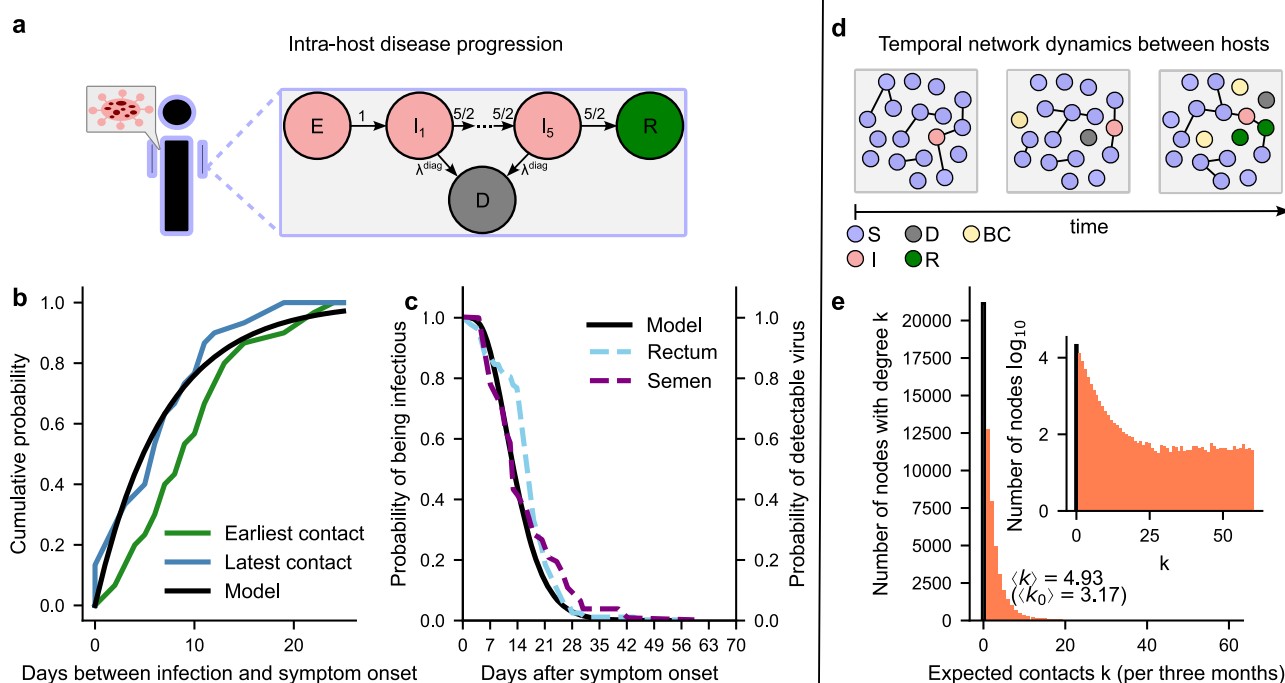

**Fig. 1 | Components of the infection model and related distributions. a** Infected agents develop symptoms after an incubation period. Following symptom development, they recover by passing through five infectious compartments, during which they may be diagnosed at a rate $\lambda^{\text{diag}}$. **b** Comparison of the distribution of incubation periods from[76] with the distribution used in the model. **c** Comparison of the distribution of detectable virus in rectum and semen, based on data from[77], with the infection probability distribution used in the model. **d** Inter-host dynamics are modeled by a temporal adaptive network, where contact addition and removal

follow Poisson processes. The disease state influences contact patterns: diagnosed agents are removed until recovery, and contacts are spontaneously reduced (behavior change BC) to prevent infection. S, I, D, R denote the susceptible, infectious, diagnosed and recovered compartment, respectively. **e** Distribution of expected contacts over a three-month period. The inset shows the contact distribution on a logarithmic scale (base 10). The average degree distribution is 3.17 contacts per three months, for the active population (without zero contacts) the average is 4.93 per three months.

contacts, consequently decreasing their connection formation rate, $\lambda_i^+$, at a rate $\lambda^{\text{bc}}$, the behavior change rate. When agent $i$ alters its behavior, its contact rate is adjusted to $\lambda_i^+ u$, where $u$ is a uniformly distributed random variable within the range $[0, b]$. Here, $b < 1$ is a free parameter that governs the extent of behavior reduction, alongside the behavior change rate $\lambda^{\text{bc}}$.

Agents with reduced contacts eventually revert to their usual behavior based on their vaccination status. Vaccinated agents resume to their normal contact rate at a rate $\lambda^{\text{return vacc}}$, whereas unvaccinated agents do so at a rate $\lambda^{\text{return}}$. These return rates are derived from survey data (see Supplementary Fig. 6). Agents may also revert to their baseline behavior after receiving a mpox diagnosis and completing recovery.

## Model calibration

There are five free parameters: the infection rate $\lambda^{\text{inf}}$, diagnosis rate $\lambda^{\text{diag}}$, initial size of infection $i_0$, behavior change rate $\lambda^{\text{bc}}$ and the magnitude of behavior change $b$. Both the infection rate and the diagnosis rate are derived from corresponding probabilities of infection and diagnosis, with theoretical values spanning from 0 to 1. Similarly, the magnitude of behavior change is constrained between 0 and 1, reflecting the proportionate reduction in transmission risk due to behavioral modifications. The behavior change rate $\lambda^{\text{bc}}$ represents the timescale over which behavior adapts on average, and is parameterized between hours and weeks. The initial size of infection $i_0$ is a discrete variable with possible values ranging from 5 to 20 individuals, representing the estimated number of initial cases at the onset of the modeled scenario. We denote this set of parameters by $\theta \in \mathbb{R}^5$. We model the temporal progression of diagnosed cases using a trajectory $T(\theta)$, which is sampled via the HAS framework[30]. This trajectory spans a 26-week period, providing a weekly estimate of diagnosed cases, starting on 2022-05-02. To assess the model's goodness of fit against observed data, we employ a

likelihood function defined as follows:

$$L(D|T(\theta)) = \prod_{i=0}^{25} \text{Pois}(D_i, T_i(\theta)), \tag{1}$$

where $D_i$ represents the observed number of diagnosed cases in week $i$, and $\text{Pois}(k, \lambda)$ is the Poisson likelihood function, which quantifies the probability of observing $k$ cases when the expected number of cases is $\lambda$. Here, each $T_i(\theta)$ corresponds to the number of diagnosed cases in week $i$ as determined by the sampled trajectory. This likelihood-based approach enables us to quantitatively evaluate the alignment of the model's predictions with the empirical data.

To determine the optimal set of parameters $\Theta$ that maximize the likelihood function, we implemented an Approximate Bayesian Computing based on Sequential Monte Carlo (ABC-SMC) calibration process with three steps. We used the log-likelihood function Eq.(1) as the score function. In the first step, we conducted a broad exploration of the parameter space by randomly sampling 100,000 parameter sets. In the second step, we sampled again 100,000 parameters from selected parameters from the first step and applied Gaussian perturbations. This was repeated in the third step with 500,000 parameter samples. We set the acceptance threshold of 8 times, 5 times and 3 times the log-likelihood of a perfect fit to the observed data $\log(L(D|D))$ for the first, second and third step, respectively. Ultimately, 143 trajectories were selected, each representing a parameter set whose log-likelihood was within three times the log-likelihood of a perfect fit.

## Estimation of effective reproduction number

The effective reproduction number is determined by multiplying the average number of contacts per infected individual during their infectious

period by the infection probability. We estimated the infectious period to be $t_{inf} = 2$ (weeks). The expected number of new contacts $\mu_i(t)$ during an interval of length $t$ for agent $i$ can be calculated as[30]

$$\mu_i(t) = \sum_{j \neq i} 1 - e^{-\lambda_{ij}^+ t}. \tag{2}$$

Let $p_{inf}$ denote the infection probability used in the simulation, the effective reproductive number is estimated by

$$R_t = p_{inf} \sum_{i \in S(t)} \mu_i(t_{inf}). \tag{3}$$

## Alternative scenarios

In addition to our primary model, we calibrated four alternative models, each based on slightly different assumptions to explore their impact on mpox transmission dynamics and sensitivity of the model to delays in reporting. In the first scenario, no vaccinations were administered. The second scenario assumed that agents adjusted their behavior only upon receiving a diagnosis, rather than in anticipation of an infection ($\lambda^{bc} = 0$). The third scenario restricted behavior change to only those agents who indicated a strong behavioral change in the survey, thereby limiting the pool of agents eligible to reduce their contacts during the simulation. However, the calibration for both the second and third scenarios was not successful. In the final calibration step, these scenarios yielded only trajectories with likelihoods within five times the likelihood of a perfect fit, indicating suboptimal alignment with the observed data. In addition, we calibrated the model to an adjusted epidemic curve which reflects reporting delays.

## Imports on the immunized network

To assess the immunity acquired by both infection and vaccination, we simulated the re-introduction of mpox into the network. In this scenario, 7745 second doses of the vaccine were distributed to agents who had already received their first dose. Vaccine efficacies were varied, with first doses ranging from 64% to 88% and second doses from 72% to 92%[31]. We sampled 100 combinations of vaccine and infection efficacies, ensuring that the efficacy of the first dose was less than that of the second dose, which in turn was less than the immunity acquired through infection[32]. For each efficacy combination, we conducted 10 stochastic simulations across each of the 143 networks produced in the final calibration step. For immunized agents, the infection probability was multiplied by 1 minus the protective efficacy of the source of immunization. The infection rate was again calculated from the infection probability. Agents adapted their behavior during these simulations in the same manner as they did during the 2022 outbreak. However, agents who received a vaccination or a clinical diagnosis of mpox during the simulation of the 2022 outbreak maintained their behavior without further changes, even if they had altered it during the initial outbreak. Behavior changes were initiated after ten clinically diagnosed cases were reported in the Berlin contact network.

Additionally, we conducted simulations under two different conditions: with a vaccine efficacy of 0% (providing immunity solely through infection), and without infection-conferred immunity (where protection was derived only from vaccines). These additional simulations allowed us to isolate the impacts of behavior, vaccination, and infection-driven immunity on the mpox transmission dynamics.

To account for demographic changes, we simulated two additional scenarios: one where the contact behavior of a subset of the population is shuffled, and another where a subset of the population is replaced by naive agents. In all simulations, the distribution of expected contacts remains consistent. The subpopulations are randomly sampled at the start of each simulation. For each set of parameters, a total of 14,300 simulations are conducted, representing all combinations of vaccine efficacy and the calibrated networks.

## Results

### Coupled within- and between host viral dynamics

We modeled the within-host dynamics of viral shedding using a descriptive model (Fig. 1a), akin to refs. 29,33. This model was parameterized using secondary case data from households, and describes the incubation time (which we equated to time to infectiousness, Fig. 1b), as well as the duration of viral shedding, (Fig. 1c). We then integrated this within-host model of virus shedding into a temporal contact network[34,35] to model the mpox outbreak within the Berlin gay population. In the epidemiological model, transmission-relevant contacts between individuals (agents) evolve dynamically, also in response to risk-averting behavior (Fig. 1d). We used survey data to initialize agent characteristics, including the number of sexual contacts (Fig. 1e), vaccination status and behavioral responses during the 2022 mpox outbreak[28]. Behavior changes reported by individuals were modeled by correspondingly adjusted contact rates, representing the rate and magnitude of behavior modification. In addition, the model accounted for agents' return to baseline behavior, dependent on vaccination status and recovery. In the model, vaccinations were distributed weekly according to the mpox vaccination timeline (see Supplementary Fig. 1). We assumed 80% efficacy (mpox risk reduction) among susceptible agents post-vaccination[25]. This comprehensive modeling approach enabled us to investigate the impact of contact dynamics, vaccination, and behavior changes on mpox transmission and epidemiology.

### Dynamics of the 2022 outbreak

While we were able to assign almost all parameters of the model as outlined above, the infection probability per contact, the case ascertainment (diagnosis) probability, as well as the number of initial imports into Berlin were unknown. Henceforth, we utilized a Bayesian approach (see "Methods") to fit these remaining parameters utilizing epidemiological data from the 2022 mpox outbreak (Fig. 2a). The posterior distribution of the infection probability had a 95% credible interval (CrI) of 65–70%, see Fig. 2b. This interval reflects the likelihood of infection transmission occurring during a contact with an infected individual before the connection is dissolved. The 95% CrI of the diagnosis probability was estimated to be 22–28%, meaning that approximately one in four cases was diagnosed before recovery and entered the reporting system. We estimated the initial size of the Berlin outbreak to be between 11 and 17 imports with 95% probability, Fig. 2b.

Based on the parameterized model, our simulations indicate that, on average, 12% (95% PI: 10–13%) of gay men in Berlin became infected with mpox, and 2.81% (95% PI: 3.04–3.26%) were clinically diagnosed by the end of the outbreak (Fig. 2c). At that time, a total of 37% (95% PI: 36–38%) of the population was immunized, either through infection or by receiving at least one vaccine dose (Fig. 2c). Additionally, 22% of the population spontaneously reduced their behaviors during the simulation. Based on the model, we estimated that the basic reproduction number $R_0$ was 2.13 (95% prediction interval, PI: 1.19–3.13) at the onset of the outbreak. By the second week, the effective reproduction number $R_t$ rose to an average of 2.43 (95% PI: 1.91–2.93) before it began to decline (Fig. 3a). During calendar week 26 (2022-06-27 to 2022-07-03), $R_t$ reached the critical threshold of 1, the point at which the outbreak began to subside. Following this week, the number of diagnosed cases decreased consistently each week. Using the standard relation to the basic reproduction number, the theoretical herd immunity threshold was estimated to be 53% (95% PI: 16–68%) of the population.

In our simulations, the share of realized contacts declined steadily until calendar week 25, at which point an average of 51% (95% PI: 49–58%) of the expected contacts prior to the outbreak were realized (Fig. 3a). Following this week, participation in the contact network began to increase, reaching an average participation level of 72% (95% PI: 70–76%) by the end of the simulation. This rise in contact participation was attributed to the introduction of the vaccine, with the first doses being administered in calendar week 25. The minimum network participation coincided with the effective reproduction number $R_t$ reaching the critical threshold of 1, marking a turning point in the outbreak dynamics (Fig. 3a). Subsequent increases in participation primarily involved contacts with recovered agents, who are no

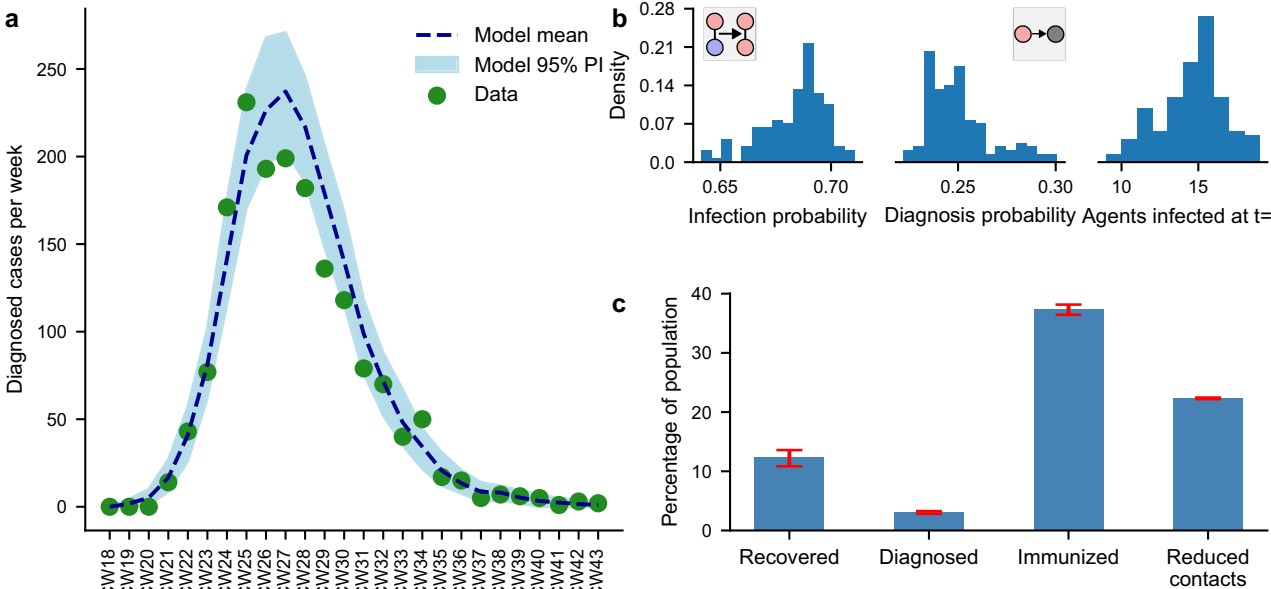

**Fig. 2 | Overview of epidemic dynamics and population response. a** Epidemic curve presented per calendar week (CW) in 2022, contrasted with the model mean and the 95% PI. **b** Posterior distribution of epidemic parameters. **c** Mean proportion of the population that became infected, diagnosed, or immunized (either through infection or vaccination) throughout the simulation, as well as the proportion that reduced their contacts at any point during the study period. Red error bar denotes the 95% PI based on 143 simulations.

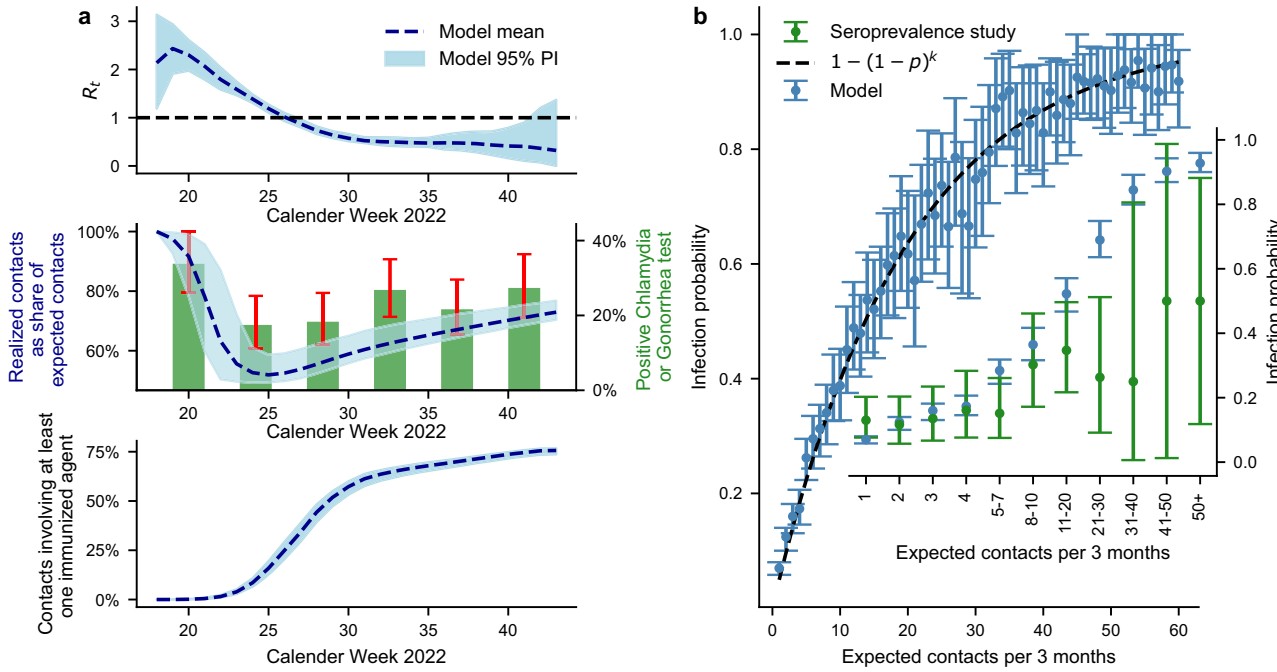

**Fig. 3 | Dynamics of the effective reproduction number $R_t$, realized contacts, immunized contacts over time, and infection probability relative to expected number of contacts. a** This plot illustrates the effective reproduction number $R_t$, the percentage of realized contacts as a share of expected contacts, and the proportion of contacts involving at least one immunized agent (through prior infection or vaccination) throughout the simulation period. The dotted line represents the model mean, while the shaded area indicates the 95% PI. The green bars represent the monthly mean percentage of positive Chlamydia or Gonorrhoea tests performed among MSM at three Berlin community-based voluntary counselling and testing centres[26] (the calendar week containing the 15th of the month as midpoints for the bars) reproduced from[27]. Data was shifted to the previous month to account for incubation periods and reporting delays. The red error bars denote 95% CI calculated using Wilson's method. **b** The main plot displays the infection probability as a function of the expected number of contacts, with dots indicating the model mean and 95% PI. The dotted line represents a binomial fit to the data, where the probability of an agent becoming infected is expressed as the inverse of the probability of never being infected: $1-(1-p)^k$, with $k$ representing the number of contacts and $p$ is estimated to be 0.04936565. The inset depicts the mean infection probability with 95% PI for contact categories compared to the results of the seroprevalence study[28]. The 95% CI for the seroprevalence study were calculated using Wilson's method. The number of participants per contact category in ascending order are 123, 86, 74, 56, 66, 43, 49, 19, 4, 2 and 6.

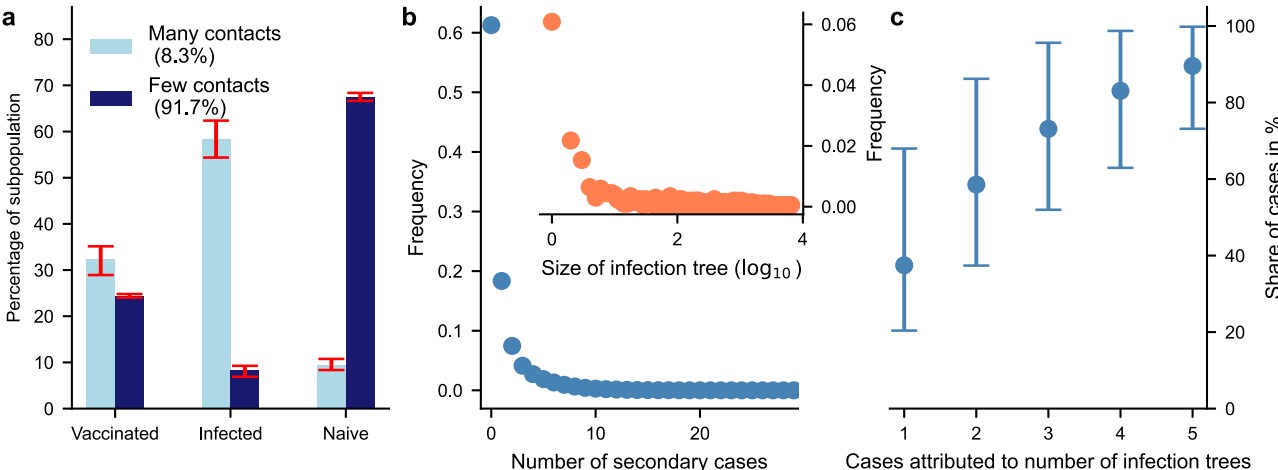

**Fig. 4 | Comparison of vaccination and infection dynamics. a** The mean proportion of vaccinated, infected, and naive individuals within the high contact group (two or more contacts in three weeks, depicted in light blue) and the low contact group (less than two contacts in three weeks, depicted in dark blue). The error bars show the 95% PI, which are based on 143 simulations. Agents who are both infected and vaccinated are counted as infected. **b** Distribution of the number of secondary cases per infected individual, represented by blue dots; and size of infection trees, indicating the number of cases traceable to a single importation event, represented by orange dots. **c** Cumulative share of cases traceable to the top imports. The plot illustrates the mean proportion of cases attributed to the import with the most cases, followed by the combined share from the two top imports, and continues in this pattern. Error bars represent the 95% PI, which are based on 143 simulations.

longer susceptible to infection (Fig. 3a). By the end of the simulation, 75% (95% PI: 73–76%) of the realized contacts involved at least one immunized agent, either through previous infection or vaccination. This trend resulted from high infection and vaccination rates among individuals with higher contact frequencies, as well as the fact that the contact network had not yet fully returned to baseline, Fig. 3a.

In our simulations, the infection probability of an individual (agent) over the course of the outbreak was strongly associated with the expected number of contacts of an individual (Fig. 3b) via a binomial model. We compared these model predictions with data from the mpox seroprevalence study[28], showing overall reasonable agreement, albeit large uncertainty ranges for high-contact individuals due to low sample sizes in this group in the sero-prevalence study.

By the end of the simulation on average 58% (95% PI: 54–62%) of individuals with at least two partners within the last 3 weeks became infected (Fig. 4a). This infection rate was considerably higher compared to the subpopulation with fewer than two contacts in the last 3 weeks, where the infection rate was just 8% (95% PI: 6–9%). In the noninfected high contact group on average 32% (95% PI: 28–35%) received a first vaccine dose, which means that on average 90% of the high contact group received some sort of immunization (Fig. 4a). In contrast, 67% (95% PI: 66–68%) of the low contact group remained naive to infection. Although the vaccination rates were of similar magnitude between the two groups (32% for the high contact group versus 24% for the low contact group), the low contact group experienced substantially fewer infections (Fig. 4a).

The number of secondary cases followed an exponential distribution, with values ranging from 0 to 29 (Fig. 4b). Most infected agents did not transmit the disease (61%), while 18% transmitted to just one person. The outbreak's persistence relied on 21% of the simulated individuals, who caused two or more secondary cases. On average, 14 initial cases were needed (95% CrI: 10–18.5, Fig. 2b), but most of these cases resulted in no or only a few secondary infections (Fig. 4b). In the majority of simulations, nearly all infections can be traced back to a few importation events (Fig. 4c). In our simulations, 73% of all cases (95% PI: 51–95%) could be traced back to three or fewer founder cases (Fig. 4c).

**Protective immunity after the 2022 outbreak**

To assess the impact of infection- and vaccination-acquired immunity on mpox outbreak potential, we performed simulations where we reintroduced mpox into the partially immunized population following the 2022 simulated

outbreak dynamics. For these simulations, second vaccine doses were administered to some individuals who had received their initial vaccination. In addition, we examined a range of vaccine efficacies[31], ensuring a higher efficacy for the second dose compared to the first, and even greater efficacy for naturally acquired immunity[32]. Across each of the 143 calibrated network configurations, we ran multiple stochastic simulations to capture the variability in mpox spread and the protective effects of immunity.

With immunity gained from vaccination and infection during the 2022 outbreak, the likelihood of experiencing a new mpox outbreak in the Berlin gay community was negligible. Almost all infection chains ended with the first person. The largest outbreak reached 0.32% of the population (Fig. 5a), which was 37 times smaller compared to the simulated 2022 outbreak. In a hypothetical simulation scenario, where the population only acquired immunity through vaccination, the average outbreak size was 3.71% of the population (95% PI: 0.03–5.24%); in a simulation where immunity was solely obtained through infection, the average outbreak reached 0.06% of the population (95% PI: 0.01–0.23%).

In simulations where immunity was acquired through both vaccination and infection, 95% of simulations resulted in no cases beyond the initial imports. In contrast, this percentage drops to 54% if immunity was solely obtained by past infection, and further decreases to 3% if immunity was acquired only by vaccination. Overall, this demonstrates the synergy between infection-induced immunity, which disproportionally affects individuals with many contacts, and immunity through vaccination, which is acquired by individuals both with many and few contacts (compare Fig. 4a).

In simulations where immunity was acquired through both infection and vaccination, the basic reproduction number $R_0$ started below one (Fig. 5b). Conversely, for the other two populations, $R_0$ initially exceeded one at the start of the simulation. In the population that acquired immunity solely through infection, $R_0$ decreased rapidly, dropping below one ~2 weeks after the outbreak began. However, in the population immunized only by vaccination, $R_0$ remained slightly above one during the first half of the simulation period and decreased at a slower rate compared to the other two populations.

To incorporate demographic changes and aspects of immune waning, we simulated various degrees of contact changes within the population (Fig. 5c), as well as random replacement of subgroups with naive agents (Fig. 5d). In both scenarios, the total population size and the distribution of expected contacts remained constant. Shuffling expected contacts was able

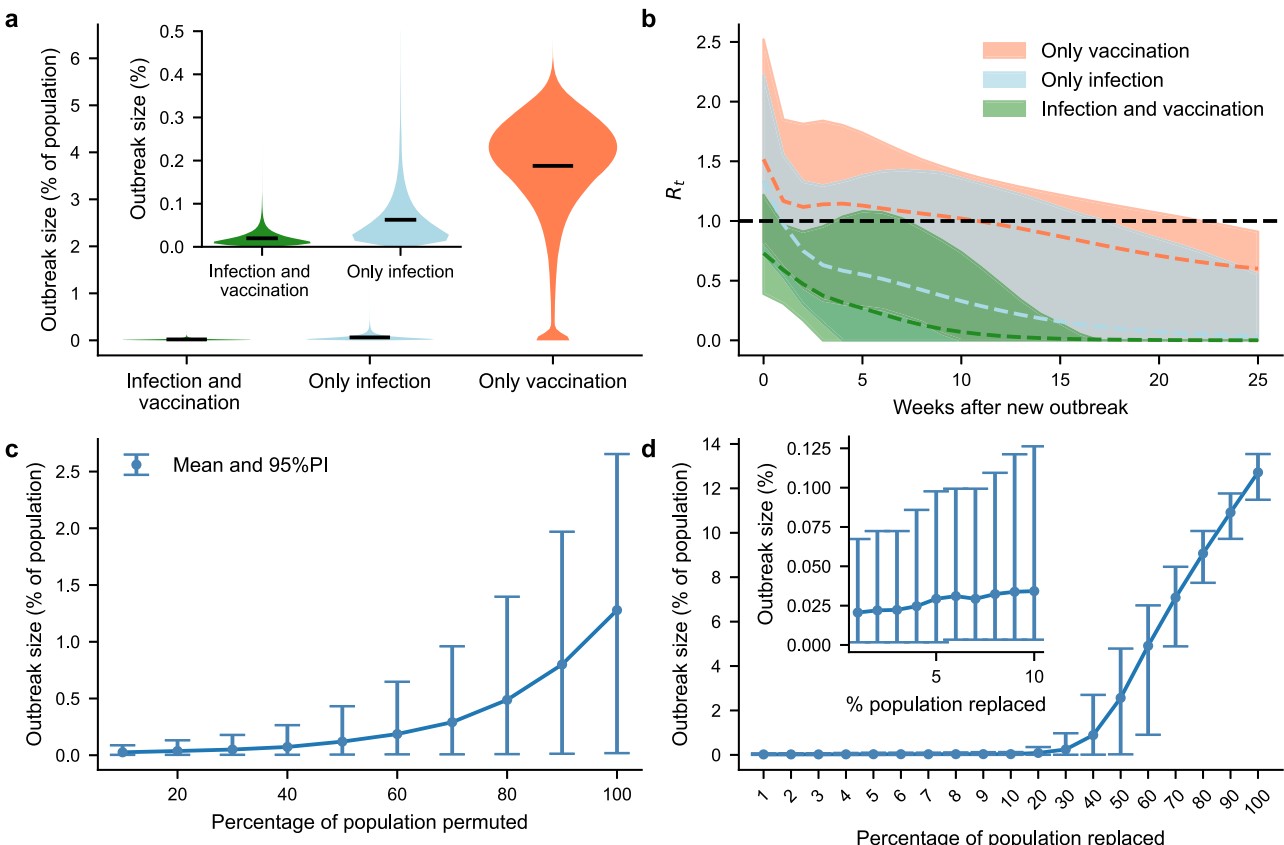

**Fig. 5 | Outbreak size and effective reproduction number for the reinfection of the immunized network. a** Distribution of outbreak sizes across 143,000 simulations for each of the three scenarios. The dark lines represent the mean outbreak size. **b** Timeline of the effective reproduction number $R_t$ for each of the three scenarios. The mean is depicted by dashed lines, and the 95% PI is shown as shaded areas. **c** Final outbreak size (mean and 95% PI) resulting from randomizing the number of contacts of a subset of the population. Each set of parameters is represented by 14,300 simulations. **d** Final outbreak size (mean and 95% PI) when a share of the population is removed and replaced with naive agents. Each set of parameters is represented by 14,300 simulations.

to increase the final outbreak size (Fig. 5c); however, even with complete shuffling, the average outbreak affected only 1.27% of the population (95% PI: 0.02–2.65%). Notably, in these scenarios, the number of immunized individuals remained constant, but in contrast to infection-acquired immunity (compare Fig. 4a), the probability of immunization became independent of the number of contacts within the network.

Replacement of immunized individuals, however, can have a significantly greater impact. When 50% of previously immunized individuals were replaced by susceptible individuals, the outbreak size surpasses the maximum effect of shuffling. With 100% replacement, the outbreak size matched that of the 2022 outbreak.

Our analyzes highlight that acquired immunity during the mpox outbreak, as well as the accompanying vaccination campaign, may have created a temporary herd immunity, that may fade over time due to demographic change (susceptibles entering the community), as well as immune waning.

## Discussion

When we parameterized our model with available viral shedding-, contact-, behavioral and epidemic data, we calibrated three parameters: We estimated a relatively high transmission probability for sexual contact (95% CrI: 65–70%), a moderate diagnosis probability (95% CrI: 22–28%) and a small number of imports (95% CrI: 11–17) that sparked the mpox outbreak in Berlin in 2022. The model-estimated number of imports is in strong agreement with contact tracing of early infections that estimated 20 cases related to travel to Spain, of which 16 attended an international pride event on Gran Canaria, before the outbreak shifted to autochthonous transmission[10]. Interestingly, our findings regarding only a few imported

cases are also consistent with other modelling studies[17], where most outbreak cases can be traced back to a small number of imports. The estimated diagnosis probability is somewhat lower than previously estimated in a modeling study comparing different countries[36]. However, under-reporting is extremely difficult to estimate from modeling incidence data alone[37], and may be higher than expected, if a proportion of individuals manifest only mild- or no symptoms of mpox[38–41]. Notably, our predictions that one in four cases was clinically diagnosed aligns well with sero-prevalence data for the Berlin MSM population[28]. A different study of the German MSM population estimates that 58% of MSM at high risk for mpox infection are immunized[11]. In this study, individuals with more than 5 contacts per calendar year are considered high risk. Our model estimates that, on average, 57% of this group is immunized, with 32% vaccinated and 25% infected (see Supplementary Fig. 7). Lastly, the estimated transmission probability for sexual contacts independently reflects values reported by other studies[42–44].

Based on the parameterized model, we studied the impact of immunization and behavioral change. Our findings indicated that vaccination had only a marginal impact on controlling the outbreak. Instead, infection-acquired immunity of high-contact individuals, combined with transient behavioral changes, played a crucial role in driving the outbreak below the epidemic threshold. Furthermore, we predicted that herd immunity was achieved by the end of the outbreak. However, demographic changes and waning immunity, as seen in other viral infections[45], are likely to erode this immunity over time, increasing the risk of future pandemics, particularly with respect to Clade I mpox, which is currently circulating in Central Africa.

The modeling indicated that transient changes in the contact network had a strong impact on the outbreak. This contact reduction could have been

influenced by both a decreased frequency of sexual encounters and heightened adoption of safer sex practices. These behavioral shifts were likely spurred by widespread information dissemination about the outbreak, both through media coverage and informal communication within the MSM population[46]. To contain the outbreak, the Berlin local health authority (LaGeSo)[47], sidekicks.berlin[48], and Deutsche Aidshilfe (German AIDS Federation)[49] and their local member organizations launched an information and behavior adaptation campaign targeting MSM in June 2022, primarily via social media and online platforms[50]. The Berlin mpox information campaign has been highlighted by the WHO as a successful example of pandemic response[50,51]. Multiple stakeholders collaborated, including health authorities, civil society, event organizers, and affected communities, which increased adherence to proposed measures, as the information came from trusted voices[51]. The campaign focused on raising awareness of risks, recognizing symptoms, and promoting measures to prevent infection and transmission[50]. According to our model, the observed behavioral changes contributed to a transient depletion of the infection-susceptible population, particularly impacting agents with at least two contacts in a 3-week period, who were the primary drivers of infection spread. Since the mean duration of infectiousness is relatively short for mpox, our simulations imply that the outbreak is sustained primarily by individuals who have at least two contacts within this period: one contact to acquire the infection and one to transmit it to another individual. On average, 58% of this highly connected subpopulation became infected. However, the temporary alterations in contact patterns resulted in a notable reduction of susceptible individuals within this group, ultimately contributing to the end of the outbreak. These findings have important public health implications. While vaccination remains the preferred strategy for primary prevention when timely, safe, and affordable, our results suggest that behavior change within key populations can occur rapidly when transmission routes and risks are clearly communicated. This underscores the need for timely, accurate, and targeted sexual health education as a crucial complement to biomedical interventions. Enabling communities to make informed choices may substantially reduce transmission, particularly in the early stages of an outbreak, when vaccine supply or uptake is limited.

Even though Orthopox infections must be reported to German health authorities within 24 h of diagnosis, there may be reporting delays in the weekly case numbers. We calibrated a model to an epidemic curve adjusted for reporting delays (see Supplementary Fig. 8) to test its sensitivity to case reporting inputs. The estimated parameters and predicted results varied negligibly from the main model (see Supplementary Fig. 9), indicating that a reporting delay of up to a week has no significant influence on the outputs of the model. The alternative models -without behavior changes or with a reduced level of behavior changes- did not accurately match the observed outbreak dynamics (see Supplementary Figs. 10–13) and neither predicted mpox seroprevalence (see Supplementary Fig. 14). In addition to external data highlighting a reduction in STI incidence (Fig. 3a), which may indirectly reflect changes in sexual contact behavior, our simulations suggest that behavior changes facilitated outbreak containment. However, even in simulations without behavior changes, the number of cases decreased over the summer of 2022, likely due to the depletion of susceptibles in the high-contact group. In this model, an average of 20% of the population became infected, with 80% of the high-contact subpopulation affected. To further delineate the impact of individual interventions on the outbreak, we also simulated alternative models that excluded vaccinations. Interestingly, these simulations suggested a minor impact of vaccinations on the dynamics of the outbreak (see Supplementary Fig. 15). Nonetheless, vaccinations may have played a crucial role in preventing a resurgence of mpox and in providing protection against severe infection. Vaccination reduces the transmissibility of mpox[52], potentially shortens the duration of virus shedding, and is generally associated with less severe disease[52,53]. For instance, infected individuals have been reported to develop fewer skin lesions[15]. Additionally, the introduction of vaccines facilitated a swifter and safer return to normal contact behaviors, thereby enhancing aspects such as quality of life within the community. For example, mpox vaccine communication was positively

associated with risky sexual practices[54], suggesting that prepandemic contact behavior more likely occurred within the vaccinated subpopulation, as implemented in our modeling.

We predicted that the likelihood of a new mpox outbreak in the Berlin MSM population was negligible by autumn 2022, whereby infection-acquired immunity played a key role. Although vaccines were relatively evenly distributed throughout the community, infections predominantly affected the high contact group and therefore impacted $R_t$ more strongly (Fig. 5b). In the scenario where both infections and vaccinations confer immunity, only 9% of the high contact group remained susceptible, which was insufficient to sustain infection chains.

Additionally, it is possible that infections occurring after vaccination or re-infection result in milder presentations of the disease[15,55], which may explain steady detection of a low number of clinically inapparent infections after 2022[27].

We predicted that herd immunity, which was acquired by autumn 2022, can erode by an influx of infection-susceptible or infection-naive individuals. In particular, significant outbreaks become more plausible when 50% immunized individuals become replaced by susceptibles. Since 2023, mpox transmission in Berlin and Germany has remained low[56]. However, MVA vaccination efforts have also declined significantly, also due to challenges in vaccine availability through pharmacies. Additionally, key drivers of vaccine uptake, such as perceived risk, personal connections to mpox cases, and mpox knowledge, are decreasing over time. Without sustained vaccination efforts for at-risk populations, there is a substantial risk that a core group capable of sustaining a future outbreak could reemerge within a few years. Although German PrEP guidelines[57] recommend MVA vaccination for individuals prescribed PrEP, the effectiveness of vaccination implementation remains unclear.

Other studies[17,18] identified either the depletion of susceptible individuals or early vaccinations[23] as the primary factor responsible for the decline of the 2022 mpox outbreak, with behavior changes deemed negligible. The first two studies[17,18] are nationwide model-based analyzes, while we concentrated on a smaller community. In both models, the populations are characterized by heavy-tailed contact distributions, but it is likely that contacts in Berlin are more evenly distributed than in a national population and participation in the network is plausibly higher in Berlin. At first glance, the predicted herd-immunity threshold of well under 1% from[18] may seem to contrast with our predictions. Firstly, even in terms of reported cases, this threshold was exceeded in MSM in Berlin, where more than 2% were diagnosed with mpox in 2022 (not accounting for undiagnosed cases). Additionally, sexual transmission networks are often geographically clustered[58,59]. As a result, the 1% of nationwide MSM required for herd immunity may translate into much higher values for clusters like Berlin or other mpox epicenters. A Canadian study reports vaccination as most successful intervention during the 2022 mpox outbreaks in Montréal, Toronto, and Vancouver[23]. The vaccination campaign in Canada started in the beginning of June 2022, 1 month before the first doses where administered in Berlin. By the middle of October 2022, the three Canadian cities had vaccination coverage of 44–58%, while Berlin has a significantly lower coverage of 25%. The authors report a decline in contacts during the 2022 mpox outbreak, but the estimates are imprecise due to unknown risk aversion already adopted during Covid-19 preventive measures[60]. In Berlin, behavior had already returned to prepandemic levels[50], highlighted by the CSD parade in June 2022, which was the first since being canceled for the previous 2 years. Interestingly, the Canadian study supports our claim that case depletion can also occur without behavior changes.

Our findings are in agreement with studies on the mpox outbreak in the UK, which indicated that behavior changes contributed to reducing cases and that vaccinations did not significantly impact on the decline of the 2022 outbreak[19,20]. Furthermore, a modeling study of the Italian mpox outbreak[21] reached a similar conclusion, attributing the decline in cases during autumn 2022 to a combination of contact reduction and transient behavior changes. Additionally, a Belgian study highlights that patients in the later stages of the pandemic engaged in less sexual risk behavior

compared to earlier phases[22]. This aligns with our observation that high-contact agents were infected first, whereas transmission dead-ends became infected at later stages.

Our model estimated a basic reproduction number of 2.13 (95% PI: 1.91–3.13) at the onset of the outbreak, which is consistent with other studies for Germany reporting values of 2.88[61] and 3.67 (90% CrI: 2.78–4.61)[36].

The statistics used in this study, including the number of gay men, distribution of contacts, and vaccination rates, were derived from the mpox seroprevalence study[28] and the EMIS-2017 study[13]. Participants for the seroprevalence study were recruited at STI/HIV clinics and checkpoints, which may bias the contact distribution towards individuals with more contacts. This group also exhibited higher vaccination rates than the general population, potentially indicating higher risk behaviors or risk awareness. To address recruiting biases, we considered vaccinated and unvaccinated agents separately. Additionally, we did not account for potential childhood smallpox vaccinations. These were mandatory worldwide until 1980 and continued in some countries. Therefore, participants over the age of 50 or those born outside Germany may have received childhood smallpox vaccinations. While these do not fully protect against mpox infection, antibodies may be detectable[62] and may partially protect from infection[63] in this group.

The model is specifically calibrated to the population of gay men in Berlin and may not generalize to other populations with differing age structure, health care infrastructure, behavioral responses, or modes of transmission. In particular, findings may not generalize to the recent mpox outbreak in Central Africa. However, the model can be reparametrized to study mpox outbreaks in MSM communities in other major European cities or to model other sexually transmissible diseases within Berlin. Given that the main component of the model is adaptive contact behavior change, simpler models may suffice in cases where behavior is not changed in response to an ongoing outbreak.

To assess the risk of a new mpox outbreak in Berlin, we simulated disease spread on the immunized network using the same infectious parameters as for the 2022 outbreak. This assumes cross-neutralization between mpox clades, as well as similar transmission routes and viral shedding dynamics. In 2024, cases of Clade Ib were reported outside Africa. While the viral shedding kinetics of Clade Ib remain unclear, it has been linked to sexual transmission[64]. With regards to immunization, Orthopoxvirus vaccines are usually effective against known mpox clades, suggesting cross-neutralization between clades, which may also translate to infection-acquired immunity. However, antibodies from vaccination or infection may wane over 3–6 months[16] and fully return to baseline 2 years post vaccination[65], making some individuals susceptible to reinfection. To study this effect, we replaced a proportion of immunized individuals with infection-susceptible (or naive) individuals in our simulations (Fig. 5d).

Interestingly, vaccination was also associated with an eightfold reduced transmissibility of mpox in a Portuguese study[52], possibly because neutralizing antibody titers are quickly generated in previously vaccinated individuals with breakthrough infection. Early induction of neutralizing vaccines could shorten the shedding of infectious virus and thus infectiousness, which is a factor that we did not consider in simulations in Fig. 5d. Moreover, it has been reported that cellular immune mechanisms may offer longer-term protection[66,67] and may also explain why vaccination is generally associated with less disease severity[52,53]. However, the Portuguese study was conducted during the second wave of mpox in 2023, a period characterized by increased vaccination coverage and potentially altered behavioral patterns or public health responses, all of which could have influenced the observed effects.

## Conclusion
Spreading of infectious diseases is driven by an intricate interplay between biological factors on the one hand, such as viral shedding kinetics, transmissibility[33] and susceptibility to infection[45], as well as social factors such as contact dynamics and the contact degree distribution[68,69]. Epidemiological modeling approaches rarely combine both social dynamics and biological factors. Herein, we introduced an integrated modeling framework to study the 2022 mpox outbreak in the Berlin gay population, which was by far the largest outbreak within Germany (see Supplementary Fig. 16) We found that while MPXV shedding kinetics are relatively short (~2–3 weeks)[70–72] compared to the speed of the contact network, mpox required high per contact transmissibility and a highly dynamic contact network for spreading. While mpox is transmitted via skin or mucosal contact, prolonged and intense exposure likely occurs primarily during sexual contact[73]. Effective spreading above the epidemic threshold ($R_t > 1$) is only possible, if the sexual contact network involves many partners within the short time frame of viral shedding. This may explain why this contact network may be particularly vulnerable to mpox[17]. In summary, our integrated modeling approach sheds light on the intricate relationship between virus shedding kinetics, transmissibility and contact network dynamics, exemplified for the 2022 outbreak in Berlin. We found that immunization of potential super-spreaders, as well as a lowering of contact degrees through transient behavioral changes, was able to push its effective reproduction number below the epidemic threshold. The mass vaccination campaign, on the other hand, started too late to impact on the epidemic decline, but prevented mpox resurge after the initial outbreak.

## Data availability
The datasets generated during the current study, as well as all input parameters for the simulations, are available via GitHub at https://github.com/KleistLab/mPox/tree/main/results and https://github.com/KleistLab/mPox/tree/main/parameters, respectively, and via Zenodo at https://zenodo.org/records/17012304[75]. The results of the seroprevalence study[28] are excluded and can be obtained directly from the study's authors upon reasonable request. The source data for all manuscript figures is available in Supplementary Data 1.

## Code availability
Codes were written in Python 3.11.6 and are available via GitHub at https://github.com/KleistLab/mPox/tree/main and via Zenodo at https://zenodo.org/records/17012304[75].

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

## Acknowledgements

Funded by the Deutsche Forschungsgemeinschaft (DFG, German Research Foundation) under Germany's Excellence Strategy - The Berlin Mathematics Research Center MATH+ (EXC-2046/1, project ID: 390685689). The authors would like to thank the HPC Service of FUB-IT, Freie Universität Berlin, for computing time[74].

## Author contributions

N.G., U.M., and M.v.K. conceptualized the paper. D.S., J.M., A.N., and A.J.S. were involved in data curation and data provision. U.M. and M.v.K. supervised the project. N.G. and H.-Y.K. performed the analysis. N.G. wrote the first draft with help from U.M. and M.v.K. N.L., A.N., and A.J.S. provided inputs to improve the content. All authors critically reviewed the manuscript and contributed to the final draft.

## Funding

## Competing interests

The authors declare no competing interests.
