## [Transparent Peer Review file · Communications Medicine]

Behavior change and infection induced immunity led to the decline of the 2022 Mpox outbreak in Berlin

Corresponding Author: Mr Nils Gubela

Version 0:

Reviewer comments:

Reviewer #1

(Remarks to the Author)
Dear Editor,

I have reviewed the manuscript entitled "Transient behavior changes and depletion of susceptibles in high-contact groups led to case decline during 2022 mpox outbreak in MSM in Berlin". This is a well-structured and methodologically robust study addressing the dynamics behind the decline of the 2022 mpox outbreak within the MSM community in Berlin. The authors employ a robust agent-based and within-host modeling approach, integrating epidemiological, behavioral, and vaccination data.

Strengths:

- The integration of biological parameters (viral shedding, transmissibility) with social dynamics (contact networks, behavioural changes) is innovative and provides a comprehensive perspective on outbreak control.
- The study challenges the assumption that vaccination alone was the key driver in reducing mpox cases, highlighting instead the role of infection-induced immunity and transient behavioural adaptations.
- The manuscript is well-written, with a logical flow and detailed explanation of methods.
- The availability of data ensures reproducibility.

Points for Improvement:

- The model focuses specifically on the MSM population in Berlin. While this allows for accurate calibration, the findings may not be directly generalizable to other contexts with different behavioral, demographic, or healthcare dynamics. A discussion on this limitation could be expanded.
- Although the authors conclude that vaccination had a limited impact on the 2022 outbreak trajectory, the potential role of vaccination in reducing disease severity, reinfection risk, and supporting behaviour normalization is not fully explored.
- It would be useful to elaborate on the public health implications of these findings, particularly regarding how transient behaviour changes could be leveraged in future interventions.

Overall, I find this manuscript to be a valuable contribution to the understanding of mpox epidemiology and outbreak dynamics. I recommend minor revisions to address the points above, primarily to improve contextual discussion.

Reviewer #2

(Remarks to the Author)

This article sheds light on the intricate relationship between virus shedding kinetics, transmissibility and contact network dynamics. I believe that incorporating the following suggestions for revision could further enhance the quality of the article:

1. In the abstract section, the authors should explicitly highlight the broader implications of this study by clearly articulating how the identified factors contributing to the decline in MSM cases in Berlin during 2022 may inform current public health strategies or shape future epidemiological research in similar urban populations.

2. The ideas in lines 12-14 need to be supported by references.

3.The authors do not make specific recommendations for response (e.g. enhanced needle vaccination programmes or surveillance of target populations), although they mention that demographic changes and declining immunity may weaken this immunity over time, thereby increasing the risk of future pandemics.

4.Contradictions with the findings of the Canadian study are only briefly mentioned, without any in-depth analysis of the specific effects of differences in the timing of vaccination, coverage or structure of exposure networks.

5.There may be delays in the collection of case reports and vaccination data (e.g., lag from diagnosis to reporting), but this is not adjusted for in the model and may affect the accuracy of parameter estimates.

Version 1:

Reviewer comments:

Reviewer #1

(Remarks to the Author)

After reviewing the authors' rebuttal and revised manuscript, I acknowledge that the requested revisions were generally addressed in a thorough and appropriate manner. Key issues raised during peer review, such as the generalizability of findings, the role of vaccination, public health implications, and comparative discussion with other international datasets, have been incorporated into the manuscript. These additions contribute positively to the clarity and depth of the work. However, I would like to raise two technical points for further consideration:

First, regarding terminology: in the revised version, there are instances where the term "MSM" (men who have sex with men) has been replaced by "gay." I would suggest reconsidering this substitution, as the two terms reflect different concepts in epidemiological and behavioural research and are not directly interchangeable.

"MSM" is a behaviour-based classification that encompasses all men who engage in sexual activity with other men, regardless of how they self-identify in terms of sexual orientation. The term "gay," by contrast, is identity-based and may inadvertently exclude individuals who do not identify as gay but are part of the at-risk behavioural group. Given that the study examines transmission risk and case patterns, which are more accurately associated with sexual behaviour than identity, the use of "MSM" remains more appropriate and scientifically precise.

Second, in line 367, the manuscript refers to a Portuguese study reporting an 8-fold reduction in mpox transmissibility associated with vaccination [55]. I suggest clarifying that this observation was made during the second wave of the mpox outbreak, as this temporal context is relevant for interpreting the finding. It likely reflects a period with increased vaccination coverage and possibly different behavioural patterns or public health responses, which could have influenced the observed effect.

Thank you for your attention to these points.

Reviewer #2

(Remarks to the Author)

The authors have addressed my previous comments, and thus I recommend this manuscript for publication.

Version 2:

Reviewer comments:

Reviewer #1

(Remarks to the Author)

Dear Editors,

After reviewing the author's rebuttal and revised manuscript, I acknowledge that the requested revisions were generally addressed in a thorough and appropriate manner.

I therefore recommend this manuscript for publication.

Thank you very much.

Best regards,

We would like to express our gratitude to both reviewers for their insightful suggestions. We greatly appreciate the feedback, which has significantly improved the quality of the manuscript. Thank you for the time and effort invested in the thorough review of our work. Below, we provide detailed responses to each of your comments.

Reviewer 1

This is a well-structured and methodologically robust study addressing the dynamics behind the decline of the 2022 mpox outbreak within the MSM community in Berlin. The authors employ a robust agent-based and within-host modeling approach, integrating epidemiological, behavioral, and vaccination data.

Strengths:

- The integration of biological parameters (viral shedding, transmissibility) with social dynamics (contact networks, behavioural changes) is innovative and provides a comprehensive perspective on outbreak control.
- The study challenges the assumption that vaccination alone was the key driver in reducing mpox cases, highlighting instead the role of infection-induced immunity and transient behavioural adaptations.
- The manuscript is well-written, with a logical flow and detailed explanation of methods.
- The availability of data ensures reproducibility.

Points for Improvement:

- The model focuses specifically on the MSM population in Berlin. While this allows for accurate calibration, the findings may not be directly generalizable to other contexts with different behavioral, demographic, or healthcare dynamics. A discussion on this limitation could be expanded.

Author response: Thank you for highlighting this point. We agree that ensuring clarity for the reader is crucial in preventing misinterpretations of our findings. Therefore, we have added a discussion on the generalizability of the results in the limitations section (lines 339ff):

“The model is specifically calibrated to the population of gay men in Berlin and may not generalize to other populations with differing age structure, health care infrastructure, behavioral responses, or modes of transmission. In particular, findings may not generalize to the recent mpox outbreak in Central Africa. However, the model can be reparametrized to study mpox outbreaks in MSM communities in other major European cities or to model other sexually transmissible diseases within Berlin. Given that the main component of the model is adaptive contact behavior change, simpler models may suffice in cases where behavior is not changed in response to an ongoing outbreak.”

- Although the authors conclude that vaccination had a limited impact on the 2022 outbreak trajectory, the potential role of vaccination in reducing disease severity, reinfection risk, and supporting behaviour normalization is not fully explored.

Author response: We thank the reviewer for pointing this out. We added more details to the discussion of vaccination in reducing disease severity and infection risk in lines 261ff:

“Nonetheless, vaccinations may have played a crucial role in preventing a resurgence of mpox and in providing protection against severe infection. Vaccination reduces the transmissibility of mpox [54], potentially shortens the duration of virus shedding, and is generally associated with less severe disease [54, 55]. For instance, infected individuals have been reported to develop fewer skin lesions [15]. Additionally, the introduction of vaccines facilitated a swifter and safer return to normal contact behaviors, thereby enhancing aspects such as quality of life within the community. For example, mpox vaccine communication was positively associated with risky sexual practices [56], suggesting that prepandemic contact behavior more likely occurred within the vaccinated subpopulation, as implemented in our modeling.”

Also, we added a reference that studies the role of vaccination on behavior normalization, which is something explicitly accounted for in our model.

- It would be useful to elaborate on the public health implications of these findings, particularly regarding how transient behaviour changes could be leveraged in future interventions.

Author response: Thank you for this suggestion. We added details about the pandemic response in Berlin to the discussion (lines 218ff) and about transient behaviour changes for future interventions (lines 236ff):

“To contain the outbreak, the Berlin local health authority (LaGeSo) [49], sidekicks.berlin [50], and Deutsche Aidshilfe (German AIDS Federation) [51] and their local member organizations launched an information and behavior adaptation campaign targeting MSM in June 2022, primarily via social media and online platforms [52]. The Berlin mpox information campaign has

been highlighted by the WHO as a successful example of pandemic response [52, 53]. Multiple stakeholders collaborated, including health authorities, civil society, event organizers, and affected communities, which increased adherence to proposed measures, as the information came from trusted voices [53]. The campaign focused on raising awareness of risks, recognizing symptoms, and promoting measures to prevent infection and transmission [52]."

"While vaccination remains the preferred strategy for primary prevention when timely, safe, and affordable, our results suggest that behavior change within key populations can occur rapidly when transmission routes and risks are clearly communicated. This underscores the need for timely, accurate, and targeted sexual health education as a crucial complement to biomedical interventions. Enabling communities to make informed choices may substantially reduce transmission, particularly in the early stages of an outbreak, when vaccine supply or uptake is limited."

Overall, I find this manuscript to be a valuable contribution to the understanding of mpox epidemiology and outbreak dynamics. I recommend minor revisions to address the points above, primarily to improve contextual discussion.

Reviewer 2

This article sheds light on the intricate relationship between virus shedding kinetics, transmissibility and contact network dynamics. I believe that incorporating the following suggestions for revision could further enhance the quality of the article:

1. In the abstract section, the authors should explicitly highlight the broader implications of this study by clearly articulating how the identified factors contributing to the decline in MSM cases in Berlin during 2022 may inform current public health strategies or shape future epidemiological research in similar urban populations.

Author response: Thank you for this suggestion. We added this to the abstract, as well as discussion (line 236ff):

"While vaccination remains the preferred strategy for primary prevention when timely, safe, and affordable, our results suggest that behavior change within key populations can occur rapidly when transmission routes and risks are clearly communicated. This underscores the need for timely, accurate, and targeted sexual health education as a crucial complement to biomedical interventions. Enabling communities to make informed choices may substantially reduce transmission, particularly in the early stages of an outbreak, when vaccine supply or uptake is limited."

2. The ideas in lines 12-14 need to be supported by references.

Author response: Many thanks for pointing this out. We added a reference for the population claim and added a supplementary figure (Fig. S2) for the distribution of mpox cases by federal state in Germany.

3. The authors do not make specific recommendations for response (e.g. enhanced needle vaccination programmes or surveillance of target populations), although they mention that demographic changes and declining immunity may weaken this immunity over time, thereby increasing the risk of future pandemics.

Author response: Thank you for pointing this out. We discussed our findings and the related recommendations for responses in lines 283ff:

"Since 2023, mpox transmission in Berlin and Germany has remained low [59]. However, MVA vaccination efforts have also declined significantly, also due to challenges in vaccine availability through pharmacies. Additionally, key drivers of vaccine uptake, such as perceived risk, personal connections to mpox cases, and mpox knowledge, are decreasing over time. Without sustained vaccination efforts for at-risk populations, there is a substantial risk that a core group capable of sustaining a future outbreak could reemerge within a few years. Although German PrEP guidelines [60] recommend MVA vaccination for individuals prescribed PrEP, the effectiveness of vaccination implementation remains unclear."

4. Contradictions with the findings of the Canadian study are only briefly mentioned, without any in-depth analysis of the specific effects of differences in the timing of vaccination, coverage or structure of exposure networks.

Author response: Thank you for raising this concern. We added a more detailed account of the different findings to the Canadian study in the discussion (lines 303ff):

“A Canadian study reports vaccination as most successful intervention during the 2022 mpox outbreaks in Montréal, Toronto, and Vancouver [23]. The vaccination campaign in Canada started in the beginning of June 2022, one month before the first doses were administered in Berlin. By the middle of October 2022, the three Canadian cities had vaccination coverage of 44-58%, while Berlin has a significantly lower coverage of 25%. The authors report a decline in contacts during the 2022 mpox outbreak, but the estimates are imprecise due to unknown risk aversion already adopted during Covid-19 preventive measures [63]. In Berlin, behavior had already returned to prepandemic levels [52], highlighted by the CSD parade in June 2022, which was the first since being canceled for the previous two years. Interestingly, the Canadian study study supports our claim that case depletion can also occur without behavior changes.”

5. There may be delays in the collection of case reports and vaccination data (e.g., lag from diagnosis to reporting), but this is not adjusted for in the model and may affect the accuracy of parameter estimates.

Author response: Many thanks for pointing this out. There is no delay in reporting of the vaccination data, as the date of vaccine administration is recorded and reported. In Germany, cases of Orthopox virus infection must be reported to the health authorities within 24 hours, and since we aggregate cases to calendar week, the effect of reporting delays should be negligible.

For sensitivity analysis, we calibrated an alternative model using an adjusted case reporting timeline, assuming a delay of up to a week between diagnosis and reporting. This analysis yielded the same results as the original analysis, as depicted below:

We calibrated the model to the epidemic curve adjusted for reporting delays and obtained similar parameter estimates and model outputs as in the original model:

We included this sensitivity analysis in the supplementary materials (in the alternative models section) and added this to the discussion (lines 244ff):

“Even though Orthopox infections must be reported to German health authorities within 24 hours of diagnosis, there may be reporting delays in the weekly case numbers. We calibrated a model to an epidemic curve adjusted for reporting delays to test its sensitivity to case reporting inputs. The estimated parameters and predicted results varied negligibly from the main model (see Supplementary Materials), indicating that a reporting delay of up to a week has no significant influence on the outputs of the model.”

We would like to thank the reviewers again for the time and effort invested in the thorough second review of our work. Below, we provide detailed responses to the remaining comments.

Reviewer 1

After reviewing the authors' rebuttal and revised manuscript, I acknowledge that the requested revisions were generally addressed in a thorough and appropriate manner. Key issues raised during peer review, such as the generalizability of findings, the role of vaccination, public health implications, and comparative discussion with other international datasets, have been incorporated into the manuscript. These additions contribute positively to the clarity and depth of the work.

However, I would like to raise two technical points for further consideration:

First, regarding terminology: in the revised version, there are instances where the term "MSM" (men who have sex with men) has been replaced by "gay." I would suggest reconsidering this substitution, as the two terms reflect different concepts in epidemiological and behavioural research and are not directly interchangeable.

"MSM" is a behaviour-based classification that encompasses all men who engage in sexual activity with other men, regardless of how they self-identify in terms of sexual orientation. The term "gay," by contrast, is identity-based and may inadvertently exclude individuals who do not identify as gay but are part of the at-risk behavioural group. Given that the study examines transmission risk and case patterns, which are more accurately associated with sexual behaviour than identity, the use of "MSM" remains more appropriate and scientifically precise.

Author response: Thank you for your suggestion and the detailed explanation. We have incorporated various data sources, some related to the MSM community in Berlin and others specific to the gay population. To ensure accuracy, we reviewed each occurrence of "MSM" and "gay" with our epidemiological partners.

Second, in line 367, the manuscript refers to a Portuguese study reporting an 8-fold reduction in mpox transmissibility associated with vaccination [55]. I suggest clarifying that this observation was made during the second wave of the mpox outbreak, as this temporal context is relevant for interpreting the finding. It likely reflects a period with increased vaccination coverage and possibly different behavioural patterns or public health responses, which could have influenced the observed effect.

Author response: Thank you for raising this point. We added a paragraph for clarification (lines 522f):

"However, the Portuguese study was conducted during the second wave of mpox in 2023, a period characterized by increased vaccination coverage and potentially altered behavioral patterns or public health responses, all of which could have influenced the observed effects."

Thank you for your attention to these points.

Reviewer 2

The authors have addressed my previous comments, and thus I recommend this manuscript for publication.